# Computational Drug Repurposing Algorithm Targeting TRPA1 Calcium Channel as a Potential Therapeutic Solution for Multiple Sclerosis

**DOI:** 10.3390/pharmaceutics11090446

**Published:** 2019-09-02

**Authors:** Dragos Paul Mihai, George Mihai Nitulescu, George Nicolae Daniel Ion, Cosmin Ionut Ciotu, Cornel Chirita, Simona Negres

**Affiliations:** 1Faculty of Pharmacy, “Carol Davila” University of Medicine and Pharmacy, Traian Vuia 6, 020956 Bucharest, Romania; 2Center for Physiology and Pharmacology, Medical University of Vienna, Schwarzspanierstrasse 17, 1090 Vienna, Austria

**Keywords:** transient receptor potential channels, QSAR, molecular docking, data mining, drug-repurposing, neurodegeneration, demyelination, antinociception, desvenlafaxine, paliperidone, febuxostat

## Abstract

Multiple sclerosis (MS) is a chronic autoimmune disease affecting the central nervous system (CNS) through neurodegeneration and demyelination, leading to physical/cognitive disability and neurological defects. A viable target for treating MS appears to be the Transient Receptor Potential Ankyrin 1 (TRPA1) calcium channel, whose inhibition has been shown to have beneficial effects on neuroglial cells and protect against demyelination. Using computational drug discovery and data mining methods, we performed an in silico screening study combining chemical graph mining, quantitative structure–activity relationship (QSAR) modeling, and molecular docking techniques in a global prediction model in order to identify repurposable drugs as potent TRPA1 antagonists that may serve as potential treatments for MS patients. After screening the DrugBank database with the combined generated algorithm, 903 repurposable structures were selected, with 97 displaying satisfactory inhibition probabilities and pharmacokinetics. Among the top 10 most probable inhibitors of TRPA1 with good blood brain barrier (BBB) permeability, desvenlafaxine, paliperidone, and febuxostat emerged as the most promising repurposable agents for treating MS. Molecular docking studies indicated that desvenlafaxine, paliperidone, and febuxostat are likely to induce allosteric TRPA1 channel inhibition. Future in vitro and in vivo studies are needed to confirm the biological activity of the selected hit molecules.

## 1. Introduction

Multiple sclerosis (MS) is an autoimmune disease targeting the central nervous system (CNS). This complex pathology is characterized by local inflammation, demyelination, and axonal loss mediated mainly by reactive lymphocytes that enter the CNS [1]. Although MS affects a vast number of young adults worldwide, the pathogenesis underlying its development is not fully understood [2].

Several pathological mechanisms promote the generation of CNS lesions in multiple sclerosis. T lymphocytes cross the blood brain barrier (BBB) and are activated by antigen presenting cells, including B cells, macrophages, microglia, and dendritic cells, thus initiating the adaptive immune response. Moreover, B cells produce antibodies and proinflammatory cytokines, triggering inflammation in the CNS [3]. Neuroinflammation is present during neurodegeneration and demyelination, which leads to chronic microglial activation, oxidative stress, mitochondrial injury, excitotoxicity, and axonal damage [4]. Unfortunately, multiple sclerosis is a pathology that does not benefit from a vast array of therapeutic solutions, unlike other CNS diseases. Available disease modifying therapies include interferons beta 1a and 1b, peginterferon 1a, glatiramer acetate, mitoxantrone, teriflunomide, fingolimod, dimethyl fumarate, cladribine, azathioprine, cyclophosphamide, natalizumab, ocrelizumab, and alemtuzumab [5,6,7]. However, most MS treatments require frequent parenteral administration, which has a negative impact on patients’ quality of life. Natalizumab, ocrelizumab, and alemtuzumab are monoclonal antibodies with high treatment costs, prescribed only as a medication for extremely active or relapsing-remitting MS that could not be managed with first-line adequate immunomodulatory therapeutics [8,9]. The antineoplastic agents (mitoxantrone, cyclophosphamide) used as second-line medication for patients with extremely debilitating pathology and unavailable therapeutic alternatives have a vast spectrum of adverse reactions [6]. Some drugs from other therapeutic classes are used for treating MS symptomatology, such as the antidepressant amitriptyline for neuropathic pain, baclofen (miorelaxant), and gabapentin (anticonvulsivant) for muscle spasms, painkillers, antidepressants, or anxiolytics [10,11,12]. Since current small molecule drugs have limited efficacy and fail to prevent relapsing in the long run, the medical scientific community is in a continuous pursuit of developing new, accessible MS medications with high efficiency and safety profiles [13].

TRPA1 is a member of the Transient Receptor Potential (TRP) superfamily of ion channels that acts mainly as a sensor for noxious stimuli and temperatures [14]. TRPA1 is a nonselective cation channel with high permeability for Ca^2+^, which either potentiates or inactivates the chemically activated receptor [15]. TRPA1 is expressed in several tissues, such as dorsal root ganglia, rodent cortex, caudal nucleus, urinary bladder, colon innervations, and pancreatic beta cells [16,17,18,19,20,21]. Furthermore, TRPA1 is present in rodent hippocampal astrocytes, contributing to basal calcium levels regulation, inhibitory synapse efficacy, and long-term potentiation [22,23]. A recent study showed that TRPA1 genetic ablation reduced mature oligodendrocytes apoptosis in the murine cuprizone model of demyelination, concluding that astrocytic TRPA1 regulates apoptosis through mitogen-activated protein kinase pathways, transcription factor c-Jun, and expression of Bak [24].

TRPA1 is a tetrameric receptor that forms a single pore and is structurally characterized by 14–16 N-terminal ankyrin repeats, which are motifs that mediate protein–protein interactions with cytoskeletal proteins [25,26]. The transmembrane subunits contain six alpha helices, an intracellular C-terminal domain, and an intracellular N-terminal domain, the latter containing reactive lysines and cysteines [26]. Several endogenous ligands that act as activators have been identified: oxidized lipids (4-hydroxy-2-nonenal, 5,6-eposyeicosatrienoic acid, prostaglandins), nitrated lipids, small reactive oxygen species, and 7-dehydrocholesterol [27,28,29,30]. Exogenous agonists can either form covalent adducts with lysines and cysteines within the intracellular N-domain (cinnamaldehyde, allicin, acrolein) [26,31,32] or induce activation in a non-covalent manner (menthol, thymol, carvacrol, nicotine, clotrimazole, nifedipine, diclofenac) [33].

Discovery of TRPA1 antagonists can be a promising tool in treating various diseases, including neuropathic pain, inflammation, and multiple sclerosis [26,34]. In a pursuit for discovering novel pharmacological agents for pain relief, several TRPA1 antagonists with proven in vivo efficacy have been developed by the pharmaceutical industry and academia: xanthine derivatives [35,36], trichloro(sulfanyl)ethyl benzamides [37], phtalimides, and related structures [38]. The gold standard TRPA1 inhibitor A-967079 was reported to bind to the receptor through an induced fit mechanism and produces subtle changes in the binding pocket conformation [39]. A molecular docking study indicated that a series of new indazole TRPA1 antagonists inhibit the cation channel by forming non-covalent bonds with residues Ser873 and Thr874 in the transmembrane (TM) domain 5 [40], which was previously shown to play an impactful role in TRPA1 activity [41,42]. Moreover, published data revealed that residues Val951 and Glu966 are relevant for propofol induced activation of TRPA1 [43], while a xanthine derivative antagonist (HC-030031) forms a hydrogen bond with Asn855 in the TM4-TM5 helix linker [44].

Novel drug design is a timely effort of multidisciplinary teams that requires vast amounts of allocated human and material resources with a relative chance of failure. A growing preoccupation has been observed worldwide for promising, less expensive routes of drug discovery. Drug repurposing implies the exploration of new potential biological targets for molecules already approved by national drug agencies or in late stages of development as a means to speed up the process of identifying new candidates for treating diseases of high interest [45,46]. Computer-aided drug design and discovery methods have been continuously improved over the past years, working to reduce costs and time investments for the identification and optimization of new lead molecules, but also as drug repurposing tools [47,48,49]. Considering the attractiveness of such strategies for discovering novel therapeutic solutions using repurposable drug candidates [50], we performed in silico screening studies based on chemical graph mining, classification and regression quantitative structure–activity relationship (QSAR) modeling, and molecular docking techniques in order to identify approved drugs as potent TRPA1 antagonists that may serve as potential treatments for MS patients.

## 2. Materials and Methods

A screening algorithm was conceptualized aiming for the discovery of novel TRPA1 antagonists using well established in silico approaches. The proposed step-by-step algorithm is summarized in Figure 1 and consisted of combining several methods in a global estimate of the probability of strong TRPA1 inhibitory activity. The implemented methodology focused on the development of a predictive model that merges state of the art screening techniques (data mining, classification and regression QSAR modeling, and molecular docking). The identified novel potential TRPA1 inhibitors were filtered thereafter by the physicochemical properties established as predictors for a high likelihood of diffusion through the BBB.

### 2.1. Datasets Preparation

The chemical structures of known human TRPA1 inhibitors and their corresponding activity values expressed as half maximal inhibitory concentration (IC_50_, M) were acquired from ChEMBL database [51]. Using the OSIRIS DataWarrior v5.0.0 software [52], the dataset was filtered by removal of compounds with inexact values of IC_50_. The chemical structures with estimated IC_50_ values greater than 2000 nM were saved separately as decoys. Mean IC_50_ values were calculated for compounds tested in multiple activity assays, duplicate structures were merged into a single entry, and negative logarithmic values of IC_50_ (pIC_50_, M) were calculated for all compounds. Three-dimensional coordinates were generated for all retained structures using OpenBabel v2.4.1 [53]. 

Compounds serving for drug repurposing screening were downloaded from the DrugBank v5.3.13 database [54] with their respective desalted 3D coordinates. The acquired database consisted of several drug groups (human and veterinary approved, experimental, investigational, nutraceutical, withdrawn, and illicit drugs) and included no organometallic or biologic drugs. All inorganic compounds and organic structures with molecular weight lower than 100 g/mol, which consisted mainly of solvents and chemical reactants, were removed thereafter.

Constitutional, topological, electronic, geometrical, and hybrid descriptors were calculated with CDK Descriptor Calculator v1.4.8 [55] and were integrated into both datasets for future analyses. Constant descriptors were removed, and descriptive statistics for pIC_50_ and the most common molecular descriptors widely used for describing druglikeness (molecular weight, partition coefficient, hydrogen bonds donors, hydrogen bonds acceptors, polar surface area, rotatable bonds) were performed using IBM SPSS Statistics v20.0 software (IBM SPSS Statistics for Windows, Version 20.0. Armonk, NY, USA: IBM Corp).

### 2.2. TRPA1 Inhibitors Structure–Activity Relationships (SAR)

As a means to identify potential TRPA1 channels blockers among existing drugs, we explored structure–activity relationships (SAR) of the inhibitors dataset. To establish the key physicochemical features of potent antagonists, scaffold analysis was performed using DataWarrior to generate the Bemis-Murcko skeletons as well as plain and most central ring systems. Bemis-Murcko skeletons are molecular frameworks that result from the removal of atom types, bond types, and side chains and have proven to be useful in various in silico screening studies [56,57,58]. Plain ring systems are rings with removed substitution patterns, linkers, and side chains, while most central ring systems are plain rings that are located topologically closest to the center of the molecule (graph center) [59,60]. Thereafter, the dataset was divided into two sets at a time in order to compare the means of pIC_50_ values between compounds with a specific scaffold and the remaining structures using an independent sample *t*-test. The same statistical reasoning was applied for Kier-Hall smarts descriptor (khs) as an attempt to credit both certain structural skeletons and individual atom groups with high inhibition potency. Kier-Hall Smarts descriptor counts e-state relevant fragments instead of calculating the actual e-state indices [61]. Similarity/activity cliffs were generated with DataWarrior using flexophore fingerprints with a similarity threshold set at 80%. A flexophore is a 3D versatile pharmacophore descriptor calculated based on molecular flexibility, which is represented using a complete graph. The function compares vertices and edges between maximum common substructures of two descriptor graphs [62,63]. 

### 2.3. Data Mining Protocol

SAR analysis results were further used in a graph mining approach in order to retrieve drugs with structural features similar to those specific for TRPA1 inhibitors with superior biological activity. DrugBank was screened for compounds with Bemis-Murcko skeletons, plain rings, most central rings, and atom groups that were found to be characteristic for compounds that possess significantly higher pIC_50_ values among the inhibitor dataset. Secondly, previously generated flexophore descriptors were used to score drugs that feature a structural similarity with TRPA1 inhibitors above the 80% threshold, using DataWarrior to search the chemical space of DrugBank database for flexophore similarity pairs between TRPA1 antagonists and repurposable molecules [63]. A scoring function was constructed as a tool to prune the DrugBank dataset by giving one point for each skeleton, most central ring, plain ring, and atom group common to the stronger TRPA1 inhibitors and another point for a flexophore similarity higher than 80%. The same scoring function was applied to both the inhibitor and the decoy datasets using only the features common to the screened repurposing dataset.

### 2.4. Quantitative Structure-Activity Relationship (QSAR) Modeling

#### 2.4.1. Binary Classification Model

The first step in predicting TRPA1 inhibitory activity for the screened database consisted of building a binary classification model based on setting cutoff values for several descriptors, similar to our previous work [58]. The TRPA1 blockers dataset was divided into three groups that included weak inhibitors with pIC_50_ values < 6 M, moderate inhibitors with pIC_50_ ranging between 6–7 M, and strong inhibitors with pIC_50_ > 7 M. This rationale was applied in order to obtain two balanced sets of strong and weak inhibitors, which are required for building classification models with good performance [64]. Values higher than 7 M were chosen for strong inhibitors, considering that compounds with IC_50_ < 100 nM are generally accepted as potent inhibitors. Strong inhibitors were labeled as active (1) and weak inhibitors as inactive (0). An independent sample *t*-test was applied in order to identify molecular descriptors that were statistically different between weak and strong inhibitors, and a weighted index was calculated by dividing the mean difference to the range of the descriptor values between the two groups. Descriptors with weights > 0.2 and receiver operating characteristics (ROC) areas under the curve (AUC) > 0.8 were further processed by building the correlation matrix and were referred to as variables. Variables that were highly intercorrelated (*R* > 0.9) were removed, and the two inhibitor groups were randomly split into training (70%) and test (30%) subsets. Cutoff values of the classifiers were chosen using ROC curves and by identifying the coordinates with a good balance between sensitivity and specificity. The test subset was then used to validate the classification model, and all classification evaluation parameters were calculated (sensitivity, specificity, accuracy, ROC AUC, and F1 score). The classification model was applied to the DrugBank and the decoy datasets.

#### 2.4.2. Regression Model

Multiple linear regression models (MLR) were built to quantitatively predict the biological activity (pIC_50_) of screened drugs on TRPA1 calcium channel. The inhibitor dataset was randomly divided into ten training (70%) and ten test (30%) subsets by a 10-fold bootstrapping randomization. The independent variables were chosen by applying forward (FW) and stepwise (SW) selection methods. The inclusion criterion was based on more exigent values for the probability of F (*p* < 0.01 for acceptance and *p* = 0.01–0.05 for removal) in order to diminish redundancy generated by the inclusion of a large number of descriptors. The forward selection method adds descriptors progressively to the equation, weighting its ability to increase the fitness of the model, while the stepwise selection method adds each descriptor in a step-by-step manner, calculating the significance of the previously included variable and removing the already added descriptors that are no longer relevant to the fitness of the regression model [65]. Each model was used to predict the activity of the test subsets for external validation. The fittest model was chosen by the highest squared correlation coefficient (R^2^*pred*) and the lowest root mean square error (RMSE*pred*) of the test subsets and was further used to predict pIC_50_ values for both DrugBank and decoy datasets.

### 2.5. Molecular Docking Simulations

A molecular docking experiment was carried out to estimate the predicted binding affinity of screened molecules to the TRPA1 channel. The crystal structure of human TRPA1 was retrieved from RCSB Protein Data Bank (PDB code: 3J9P) [39] and was subjected to several optimization steps due to low resolution and large B-factors. Missing residues were added to the protein, and loops were built and refined with Chimera v1.13.1 [66] and Modeller v9.22 [67]. Thereafter, the protein structure was energetically minimized using the AMBER ff14SB force field for further refinement. 

Three-dimensional structures of TRPA1 inhibitors, decoy dataset, and DrugBank compounds were prepared for docking using Open Babel. Ligand structures underwent hydrogen atom addition and energy minimization with MMFF94s force field, were converted to the docking file format with protonation states corresponding to pH = 7.4, and finally were imported into the PyRx v0.8 [68], a virtual screening software that runs with AutoDock Vina v1.1.2 docking algorithm [69]. Previously reported analyses of electron density maps and molecular dynamics simulations indicated that inhibitors A-967079 and HC-030031 bind to TRPA1 and interfere with channel gating via an induced fit mechanism [39,44]. Therefore, a docking simulation with flexible residues was first employed with both gold standard inhibitors (positive controls) to generate side chain conformers appropriate for a putative TRPA1-inhibitor complex. A-967079 and HC-30031 have two different binding sites, which are separated by TM5. Residues Leu870, Ser873 (cytoplasmic), Thr874, Phe877, Ile878, Leu881 (TM5), Thr908, Phe909, Met912, Leu913 (pore helix 1), Val942, Thr945, Ile946, Val948, Ile950, Leu952, and Leu956 (TM6) were set as flexible for A-967079 docking, while Arg872 (TM5) and Asn855 (TM4-TM5 helix linker) were chosen for HC-030031. Thus, the protein structure was subsequently optimized to reflect an inhibited state by replacing the original binding pocket side chains with the new conformations. 

Following receptor structure preparation, the virtual screening was performed for all datasets with rigid residues, and the searching space (grid box) was defined to include both reported adjacent binding sites. Docking scores (binding energies, Δ*G*, (kcal/mol) corresponding to the first conformation generated for each ligand were retrieved for the screened compounds (TRPA1 inhibitors, decoys, and repositioning candidates). Graphical depictions of ligand poses and interaction diagrams were built using BIOVIA Discovery Studio Visualizer (BIOVIA, Discovery Studio Visualizer, Version 17.2.0, Dassault Systèmes, 2016, San Diego, CA, USA). Validation of the docking experiment was performed by redocking the two positive controls and calculating the squared correlation coefficient (*R*^2^) between the experimental pIC_50_ and the docking scores of the TRPA1 inhibitor dataset.

### 2.6. Ranking of Potential Novel TRPA1 Inhibitors

The final step of the screening algorithm consisted of building a ranking system to identify the drugs and the druglike compounds with the highest probability of exhibiting potent TRPA1 inhibitory activity. The predictive model was built by combining the outputs of the aforementioned screening steps into a single, global equation (Figure 1). A binary logistic regression model was generated using three independent variables: the weak/strong inhibitor class as a categorical variable, the predicted pIC_50_ (pIC_50_*pred*), and the predicted binding energy (ΔG) values as continuous variables. The TRPA1 inhibitor database was divided by 10-fold bootstrapping into random training (70%) and test (30%) sets for external validation, and ten regression equations were generated. A secondary external validation was performed by calculating the inhibition probability of the decoy compounds. The model with the highest accuracy and ROC AUC was then applied to the filtered DrugBank database to estimate the probability of blocking TRPA1 channels. Since the final aim of the screening was to discover repurposable drugs as novel potential TRPA1 blockers addressing multiple sclerosis, the compounds with a probability higher than 50% of being potent inhibitors were filtered by the physicochemical properties needed for a good BBB permeation and CNS exposure: molecular weight under 500 g/mol, AlogP ranging between 2–5, polar surface area under 90 Å^2^, 3 or fewer hydrogen bond donors, and 7 or fewer hydrogen bond acceptors [70].

## 3. Results

### 3.1. TRPA1 Inhibitors and Repurposing Datasets

A dataset composed of 576 human TRPA1 inhibitors with biological activity expressed in IC_50_ values (M) was downloaded from the ChEMBL database [51]. Following the application of filtering procedures, a virtual chemical library was built by retaining 371 compounds from the original dataset, while 76 compounds were proposed separately as decoys. Drug repurposing dataset preparation resulted in a virtual library containing 7710 drug structures retained from DrugBank [54]. For each compound library, a total of 282 molecular descriptors were calculated, and 53 constant descriptors were removed. Descriptive statistics of pIC_50_ values and druglikeness-related descriptors for the active dataset are shown in Appendix A.

### 3.2. Structure-Activity Relationships of TRPA1 Inhibitors

The scaffold analysis of TRPA1 inhibitors dataset yielded 46 Bemis-Murcko skeletons, 51 plain rings, and 26 most central rings. Several scaffolds were associated with significantly higher biological activity, and the statistical significance threshold was set to *p* < 0.01 as a means to increase the discriminant power of the test (Table 1). 

Bemis-Murcko skeletons correlated with significantly higher activity values are shown in Figure 2. Structural skeletons **BM-1** (1-[3-(3-cyclohexylcyclopentyl)propyl]-decahydronaphthalene) and **BM-3** (1-[3-(3-cyclohexylcyclopentyl)propyl]-octahydro-1*H*-indene) were highly similar, well represented among the inhibitors with comparable mean pIC_50_ values, and had several common structural features: a condensed bicyclic structure represented by either nine or ten atoms rings, a three-atom linker, and a pentacyclic substructure linked to a six-atom ring. Structural skeleton **BM-2** ([1-(4,5-dicyclohexylpentyl)cyclopentyl]cyclohexane) was less frequent, being specific for only four TRPA1 inhibitors. However, structures containing this specific scaffold had a higher inhibition potency, all compounds showing pIC_50_ values higher than 7 M.

The plain rings systems specific for the TRPA1 inhibitors were split into individual rings in order to analyze their contribution for a high biological activity and are depicted in Figure 3. It was noted that four of the eight total rings were bioisosteres of ring **PR-1** (1,2,3,4-tetrahydroquinazoline-2,4-dione), which was contained in structures with pIC_50_ values ranging between 6.5–8.5 M limits and was specific for Bemis-Murcko skeletons **BM-1** and **BM-3**. Rings **PR-2** (1*H*,2*H*,3*H*,4*H*-pyrido[2,3-d]pyrimidine-2,4-dione) and **PR-3** (1*H*,2*H*,3*H*,4*H*-thieno[2,3-d]pyrimidine-2,4-dione) were less representative, but the replacement of the phenyl aromatic ring with pyridine or thiophene gave higher mean pIC_50_ values overall. The highest increase in mean activity values could be observed in compounds containing ring **PR-5** (4*H*,5*H*,6*H*,7*H*-[1,2]thiazolo[5,4-d]pyrimidine-4,6-dione), which included a substitution of the phenyl ring with thiazole, while ring **PR-4** (1*H*,2*H*,3*H*,4*H*-furo[2,3-d]pyrimidine-2,4-dione) was found in representants with furan-including rings and pIC_50_ values higher than 8.5 M (Figure 3).

Ring **PR-6** (benzene) was widely spread among the dataset, being present in 360 structures out of 371, and had a higher mean pIC_50_ value than those that did not include it. Ring **PR-7** (cyclopentane) was specific only for compounds represented by Bemis-Murcko skeleton **BM-2**. Ring **PR-8** (1,3-thiazole) was found in a large number of TRPA1 inhibitors and was specific for Bemis-Murcko skeletons **BM-1** and **BM-3,** and the vast majority of compounds that contained it had pIC_50_ values higher than 7 M. Notably, the only most central ring that was found with a significantly higher inhibitor activity was 1,3-thiazole (**MCR-1)**, which was a substructure of the same compounds that contained ring **PR-8** as a plain ring.

Kier-Hall smarts descriptors were further used to identify relevant fragments for potent TRPA1 inhibition. Instead of using the quantitative values of the descriptor, structures were grouped for statistical analysis, taking into consideration only the presence or the absence of each specific e-state fragment. The fragments present in inhibitors with significantly higher pIC_50_ values (p < 0.01) are shown in Table 2.

Statistical analysis of the atom groups counted by Kier-Hall smarts descriptor function showed that 13 atom groups were found in structures with higher inhibition potency. Notably, six of the identified atom groups had mean pIC_50_ values higher than 7 M: a carbon atom forming four covalent bonds with different non-hydrogen atoms (**AG-4**), a nitrogen atom with one single and one double bond (**AG-5**), a substituted cyclic nitrogen (**AG-7**), a hydroxyl group (**AG-8**), a cyclic oxygen (**AG-10**), and a cyclic sulfur atom (**AG-12**). Clearly, the carbon atom that formed four covalent non-hydrogen bonds was specific to the Bemis-Murcko skeleton **BM-2** and other compounds, while the aromatic oxygen, the sulfur, and the substituted nitrogen were common for both skeletons **BM-1** and **BM-3** and plain rings **PR-1-5** and **PR-8**.

Analysis of structure similarity/activity cliffs of the TRPA1 inhibitors’ chemical space based on flexophore descriptor generated several structural clusters, which are shown in Appendix A. Only six activity cliffs could be observed, two being present in the largest generated cluster. One notable activity cliff was formed between compounds **CHEMBL3981381** and **CHEMBL3976217**, which were both included in the set containing Bemis-Murcko skeleton **BM-3**.

This individual activity cliff indicated that both the substitution of furane ring with pyrazole and the presence of a different substitution pattern of fluorine atoms on the phenyl ring produced a decrease in TRPA1 inhibition. Both compounds contained nitrogen, forming both double and single bonds, a hydroxyl group, aromatic sulfur atoms, substituted aromatic nitrogen atoms, fluorine atoms, and a thiazole substructure as the most central ring, while only compound **CHEMBL3976217** contained an aromatic oxygen atom. Another cluster containing highly active structures included compounds representative for Bemis-Murcko skeleton **BM-2** (**CHEMBL3907685**, Appendix A), which also featured hydroxyl, fluorine, and chlorine atom groups.

### 3.3. Data mining and Scoring

A graph mining strategy was further used to identify potential TRPA1 inhibitors by scanning DrugBank for drugs that feature previously established scaffolds and atom groups with statistical relevance. The number of successfully retrieved drugs by applying this method is shown in Table 3. No compounds were found using the Bemis-Murcko scaffolds, probably due to the high specificity of the selected skeletons to the TRPA1 blockers.

Six structures were found to contain **PR-1** plain ring in their molecule, three being investigational drugs (selurampanel, elinogrel, and SP-8203), two were experimental druglike compounds, and one was an approved drug (ketanserin). **PR-3** was found in only two investigational drugs within the same therapeutic group (relugolix and sufugolix). **PR-7** was specific for 21 compounds, while **PR-6** (phenyl) was identified for a large number of structures (2442), being rather common and therefore unspecific. Although 82 drugs were found to contain the **PR-8** ring (1,3-thiazole), only 46 featured this substructure as the most central ring.

As expected, **AG-1** atom group was present among a high number of the screened structures, since it is highly unspecific. Atom groups **AG-5** (single and double-bonded hydrogen), **AG-10** (aromatic oxygen), **AG-12** (aromatic sulfur), and **AG-13** (chlorine) were found in fewer than 1000 screened compounds.

The data mining criterion based on flexophore descriptor similarity generated 981 pairs between 203 known TRPA1 inhibitors and 356 DrugBank entries. Similarity percentages ranged from 0.8101 to 0.9832, and the top compound pairs are presented in Appendix A. Six pairs were found with a similarity higher than 95%, five being druglike experimental compounds with no clinical usage, while melatonin (**DB01065**) was found to have a flexophore similarity of 95.33% with TRPA1 inhibitor **CHEMBL3297780** (Appendix A).

An analysis of similarity/activity cliffs was performed again using flexophore descriptors after merging the TRPA1 inhibitors dataset with the retrieved drugs with over 80% similarity. Most of the screened drugs formed a similarity network between TRPA1 inhibitors clusters, while 21 compounds formed no links within the network (Appendix A). However, the cluster with the highest density of strong inhibitors formed no direct links with the screened drugs.

A data mining scoring function was implemented in the screening protocol using the structural scaffolds and the atom groups with statistical significance to the TRPA1 inhibitory activity, awarding one point for each feature. Flexophore similarity above the 80% threshold was also awarded one point, and the total sum of all positive features was calculated as the data mining score for the screened DrugBank database, TRPA1 inhibitors, and decoys, respectively. The calculated scores summed the presence of similarity, **MCR-1** as most central ring, plain rings **PR-1**, **PR-3**, **PR-6-8,** and the atom groups with mean pIC_50_ values above 7 M (**AG-4**, **AG-5**, **AG-7**, **AG-8**, **AG-10**, **AG-12**). The score ranged from 1–10 (4.82 ± 2.82) for the TRPA1 inhibitor dataset, 0–6 (1.50 ± 0.92) for DrugBank library, and 1–4 (2.41 ± 0.79) for the decoy dataset.

DrugBank entries with either high flexophore similarity and a minimum of two common graph features or no similarity and at least three common graph features were selected for further screening, the inclusion criteria being set to a threshold value equal to three. Thus, a total of 903 repurposable structures were selected.

### 3.4. QSAR Models

#### 3.4.1. Activity Class Prediction

A classification model was built by performing ROC analysis of molecular descriptors generated for the TRPA1 inhibitors. Out of the 229 available descriptors, 188 showed a statistically significant difference between the two inhibitor classes. However, 104 had calculated weights higher than 0.2, and only 60 had ROC AUC values over 0.8. After the removal of highly intercorrelated variables, only four descriptors were selected for the classification model (Table 4). The classification model built showed that strong TRPA1 inhibitors were characterized by a high difference between charge weighted partial positive surface area and charge weighted partial negative surface area, a high eccentric connectivity index that is a highly discriminative topological descriptor that combines distance and adjacency information [71], superior order six Kier-Hall Chi path index, and superior order four Kier-Hall Chi path cluster index.

The generated classification model showed substantial statistical parameters with good accuracy for both training (calibration) and test (validation) datasets and an overall prediction accuracy of the activity class of 83.5% as well as an ROC AUC of 0.890 (Figure 4).

The classification evaluation metrics are shown in Appendix A. The true negative rate (selectivity or specificity) was higher than the true positive (recall or sensitivity) rate for both sets, the model having a higher chance of detecting the true negative compounds and a slightly lower chance of detecting the true positives. The application of the classification model to the selected repurposable drugs yielded 424 potentially active hits out of 903 screened structures.

#### 3.4.2. Quantitative TRPA1 Antagonist Activity Prediction

The quantitative biological activity, expressed as the predicted negative logarithmic value of IC_50_ (pIC_50_*pred*) of the repurposing library against the TRPA1 channel, was calculated using MLR methods. Ten stepwise and ten forward equations were built, and one model generated by both methods was chosen based on the model quality parameters. The regression coefficients are presented in Equation (1).
(1)pIC50=0.0004×GRAV4−0.3038×khs.dO−0.0954×nHBAcc+0.2523×C2SP3+0.0117×DPSA3−0.0944×khs.ssCH2−1.1296×nAcid+1.2054×khs.ddsN−0.0600×C2SP2−3.2101×MDEN13+0.0939×C3SP2+4.2769.

The generated QSAR models showed R^2^ parameters between 0.671–0.748, R^2^*pred* values ranging between 0.499–0.681, while residues varied between −2.37–2.19. The selected model was identified by both forward and stepwise independent variable selection methods, and the included descriptors are reported in Table 5.

Validation of the QSAR regression model was successful, considering that squared correlation coefficient values for both training and test subsets were higher than the threshold of 0.50, which is considered an index of acceptable fitness (Figure 5). However, the seemingly low squared correlation coefficient could be attributed to the high structural diversity of the TRPA1 chemical space, since 46 different Bemis-Murcko skeletons, 51 plain rings, and 26 most central rings were identified amongst the dataset, as reported in the SAR section. The QSAR regression equation revealed that biological activity against TRPA1 was directly proportional with the gravitational index of all heavy atoms, singly bound carbons bound to two other carbons, the difference between charge weighted partial positive surface area and charge weighted partial negative surface area, number of nitro- group e-state fragments, and doubly bound carbons bound to three other carbons. Moreover, the number of keto oxygen e-state fragments, the number of hydrogen bond acceptors, the number of linker carbon atoms, the doubly bound carbons bound to two other carbons, and the molecular distance edge between all primary and tertiary nitrogen atoms were negatively contributing to the predicted measure of biological activity. The selected regression equation was thereafter used to predict pIC_50_ values for the drug repurposing library. The mean estimated pIC_50_ values were 5.99 ± 1.31 M, varying from 0.78 M to 10.89 M. Out of 903 screened compounds, 176 molecules showed predicted pIC_50_ above 7 M.

### 3.5. Molecular Docking

A molecular docking screening study was conducted as a tool to predict the binding affinities of the screened ligands. Docking using a flexible residues approach generated favorable conformations into the previously reported binding sites for both A-967079 and HC-030031 (Appendix A). The simulated protein-ligand complex of A-967079 revealed that the TRPA1 inhibitor formed a hydrogen bond with key TM5 residue Thr874 via the oxime moiety and participated in halogen interactions with Val948 and Met912 and in other weak interactions with the binding site side chains (Appendix A). Moreover, HC-030031 interacted with the specific binding site by forming hydrogen bonds with Asn855 (TM4-TM5 helix linker), Arg872 (TM5), Arg975, and Gln1031 and participated in several weak interactions (Appendix A). Redocking both ligands with rigid residues and with a grid box simultaneously containing both binding sites yielded similar results. Thus, the favorable conformers of residues situated in both binding pockets were used in the following virtual screening protocol.

The docking scores (ΔG) of TRPA1 inhibitors ranged from −9.7 to −4.9 kcal/mol with a mean value of −7.57 ± 0.89 kcal/mol. A low squared correlation coefficient between experimental pIC_50_ and docking scores was obtained (Figure 6, *R*^2^ = 0.226), but with strong statistical significance (Pearson test, *p* < 0.0001). Previous studies concluded that the state of the art molecular docking algorithms can properly differentiate active compounds from decoys, but the scoring functions are not entirely reliable for lead optimization, since low correlations were found between the docking scores and the experimental activity values for multiple ligand databases [72]. Moreover, statistically different mean Δ*G* values between the weak and the strong inhibitor groups (*t*-test, *p* < 0.0001) were observed, denoting that the docking algorithm had discriminative capabilities between active and inactive molecules and could be suited for high throughput screening of novel candidates. The predicted binding affinities of DrugBank screened compounds ranged from −10.4 to −4.1 kcal/mol with a mean value of −7.30 ± 1.01 kcal/mol.

### 3.6. Ranking of Potential novel TRPA1 Inhibitors

A multivariate binary logistic regression model was built as a means to rank the screened DrugBank molecules by the probability of exhibiting strong TRPA1 antagonism. The activity class code, the predicted pIC_50_, and the docking scores of weak and strong TRPA1 inhibitors were used as regression independent variables for the generation and the validation of a predictive model for the overall estimate of biological activity. The resulted model yielded a highly satisfactory accuracy of 94.3%, showing an increased predictive power of inhibitory activity for the validation subset (Appendix A). The generated logistic regression mathematical model is expressed in Equation (2).

Excellent ROC AUC values were obtained for both the calibration and the validation subset, showing a global performance value of 0.978 (Figure 7). Furthermore, a high accuracy of 92.1% was obtained using the decoy dataset as a secondary validation method; only six out of 76 molecules were incorrectly predicted as active. Thus, the proposed algorithm increased the accuracy of standalone predictive models, diminishing some of their weak points, such as low *R*^2^ values and classification accuracy below 90%.

The binary logistic regression in Equation (2) offers good insight into the influence of each variable on the probability of potent TRPA1 inhibition (P). The regression equation coefficients showed that the highest predictive weight was given to the estimated pIC_50_ values. Molecules with predicted pIC_50_ values below 5 M had an overall inhibition probability lower than 0.05%. If a candidate had a predicted pIC_50_ equal to 7 M but was classified as inactive and had a predicted binding energy equal to −6 kcal/mol, there was a 55.26% probability of TRPA1 inhibition. Moreover, if the same compound was classified as active, the probability increased to 94.96% and to 96.58% if the predicted binding energy equaled −7 kcal/mol. Therefore, the highest impact on inhibition probability was given by predicted pIC_50_, followed by the activity class and the docking score.
(2)P=11+exp[−(2.725×Class+3.967×pIC50pred−0.404×ΔG−29.986)].

Plotting the probability values against estimated binding energies of all TRPA1 inhibitors (Appendix A) revealed that nine inhibitors with over 50% inhibition probability were characterized by binding energies varying from −7 to −6 kcal/mol, while the rest of the strong inhibitors exhibited binding energies below −7 kcal/mol.

The predicted activity class, pIC_50_*pred*, and the predicted binding energy values of screened DrugBank molecules were fit into the generalized prediction model based on the previously generated logistic regression equation, and the primary ranking of the potential antagonists was performed. Thus, 310 drugs and druglike compounds registered over 50% probability of TRPA1 inhibitory activity, 196 compounds had over 90% probability, 160 showed over 95% probability, and 107 showed over 99% probability of presenting antagonist activity. The top 10 ranked molecules, their generic names, and their proven biological activities are shown in Appendix A.

The hit compounds resulted from the primary ranking system were further filtered by removing the structures with physicochemical properties placed outside the well-established cutoff values for a good blood brain barrier diffusion. Therefore, only 97 candidates (*P* > 0.5) were identified as potential TRPA1 antagonists with possible use in treating multiple sclerosis. The secondary and the final rankings of the 10 best commercially available candidates as proposed therapeutic solutions are reported in Table 6. After performing the secondary ranking, only laropiprant was found in both the primary and the secondary ranking outputs. Two approved NSAIDs, flufenamic acid, and tiaprofenic acid showed high probabilities of inhibiting TRPA1 calcium channels. However, previous studies showed that flufenamic acid both activates TRPA1 channels and blocks TRPC5, TRPM2, TRPM3, and TRPV4 calcium channels [73,74]. One limitation of the proposed screening algorithm is the apparently reduced predictive capacity of selective detection of TRPA1 inhibitors and even a possibly low ability to discriminate between agonists and antagonists.

The more interesting hit molecules discovered by the combined screening algorithm were three marketed drugs (Appendix A), two being CNS-acting molecules and one non-CNS acting drug: desvenlafaxine, the main metabolite of venlafaxine (SNRI antidepressant), paliperidone, the active metabolite of risperidone (antipsychotic dopamine D_2_ receptor antagonist and serotonin 5-HT_2A_ receptor antagonist), febuxostat (xanthine-oxidase inhibitor for gout treatment). All three molecules presented several structural features identified during SAR studies as being statistically relevant for potent TRPA1 inhibition.

The pharmacokinetic profile of desvenlafaxine indicates a good oral bioavailability (80%), while paliperidone is characterized by a low (28%) absolute bioavailability when formulated in extended release preparations [75,76]. However, paliperidone is also available as long-acting injectable suspension formulations [77]. Pharmacokinetic studies revealed that febuxostat has an oral bioavailability of at least 49%, while other authors report a higher value (84%) [78,79]. The effectiveness of febuxostat in murine models of multiple sclerosis proven by pharmacodynamic assays indicates the drug’s diffusion through the BBB [80,81].

Desvenlafaxine has one phenyl ring, one hydroxyl group, and a carbon atom that forms four covalent bonds with other non-hydrogen atoms (hydroxylated cyclohexyl moiety) and one carbon linker. Paliperidone presents a hydroxyl group, a keto oxygen, two aromatic nitrogens, one aromatic oxygen, one fluorine atom, and two linker carbon atoms. Moreover, paliperidone has two particular scaffolds specific to potent TRPA1 inhibitors: 9-hydroxy-2-methyl-6,7,8,9-tetrahydropyrido[1,2-a]pyrimidin-4-one, which resembles ring **PR-2** and the 1,2-benzoxazol-3-yl)piperidin-1-yl substructure, which can act as a bioisostere of ring **PR-5** (4H,5H,6H,7H-[1,2]thiazolo[5,4-d]pyrimidine-4,6-dione). Febuxostat has several molecular features similar to a few confirmed TRPA1 inhibitors, such as the thiazole plain ring, one carbon linker, and a nitrile group bound to a phenyl ring, this specific moiety being present in structural derivatives of the golden standard TRPA1 inhibitor HC-030031.

Binding site analysis of the docked poses was performed for desvenlafaxine, paliperidone, and febuxostat in order to estimate possible protein-ligand interactions that could lead to TRPA1 inhibition. The molecular docking algorithm revealed that desvenlafaxine could successfully bind to the A-967079 pocket via weak interactions such as cation-pi and stacked pi-pi interactions with Phe877 and alkyl, alkyl-pi, and van der Waals interactions with surrounding residues (Appendix A). Since no hydrogen bonds were formed, more docking poses were explored. Desvenlafaxine could also interact with HC-030031 putative binding sites, forming two hydrogen bonds with key residue Asn855 via the protonated tertiary amine and the aliphatic hydroxyl and between the aromatic hydroxyl and Cys1025. Moreover, desvenlafaxine participated in stacked pi-pi interaction with Phe1024, alkyl-pi interaction with Arg1030, and hydrophobic van der Waals interactions with five other residues, making it a potential allosteric modulator of the channel activity (Figure 8).

The docking simulations generated binding poses of paliperidone inside the binding site of HC-030031. Paliperidone formed three conventional hydrogen bonds with Gln968, Arg975, and Cys1025 and two carbon–hydrogen bonds with Glu854 and Gln1031. The binding conformation was stabilized by alkyl and alkyl-pi interactions with Ile858, Ala971, and Arg1030, lone pair pi interactions with Ile1029, cation pi interactions with Arg872, and attractive charges between protonated tertiary amine and Arg975. Paliperidone interacted with residue Asn855 via hydrophobic van der Waals interactions, but one unfavorable donor-donor interaction leading to repulsion was formed between the ligand hydroxyl group and Arg975 (Figure 9). However, such unfavorable contacts could be diminished by torsions of the hydroxyl group.

The binding pose of febuxostat showed that the ionizable acidic group participated in hydrogen bonding with Ser972 and that the thiazole ring accepted one hydrogen bond from Arg975, while the ether group formed one hydrogen bond with Asn855. Moreover, the TRPA1–febuxostat complex could be further stabilized by weak hydrophobic interactions, such as cation-pi interactions with Arg872, Arg975, and alkyl, alkyl-pi, and van der Waals interactions with several other residues (Figure 10). Thus, all three ligands showed favorable interactions with residues essential for TRPA1 channel gating modulation.

## 4. Discussion

TRPA1 is a promising therapeutic solution for MS patients, considering that knockout studies led to the discovery of a potential pathophysiological role of the channel in astrogliosis and oligodendrocyte apoptosis [24]. We established a new screening methodology by combining classic state of the art computer-aided drug discovery techniques into a predictive model that improved the performance of individual methods. Both ligand-based virtual screening and structure-based drug discovery tools were implemented in the construction of the algorithm: graph mining approaches based on inhibitor datasets, structure-activity relationships (SAR), quantitative structure-activity relationships (QSAR) using both classification and regression models, and molecular docking simulations.

The constructed model combined into a binary logistic regression the predicted activity classes and the predicted pIC_50_ and pK_i_ values and was characterized by over 90% accuracy. The model limitation was represented by susceptibility to overfitting, since both the QSAR classification and the linear regression models included one common topological descriptor. The key player in the global prediction model was the predicted pIC_50_, since it was the highest weighting independent variable. Interestingly, flufenamic acid was successfully predicted as a TRPA1 channel modulator, but in vitro studies reported that fenamates were TRPA1 agonists and antagonists of other TRP channels [73,74]. Thus, it can be speculated that the proposed screening algorithm is able to identify potential TRPA1 ligands, but the ability to discriminate between activation and inhibition activities remains unclear.

A large number of DrugBank molecules were predicted as TRPA1 inhibitors, such as alkaloids, antibiotics, and several antineoplastic agents, which have potential use in treating inflammatory and neuropathic pain. However, the repurposing of such drugs is impractical due to high toxicity and restrained utility. Structures of investigational drugs selurampanel, SP-8203, and relugolix showed common scaffolds with strong TRPA1 inhibitors, but the graph mining score disqualified the molecules from further screening. Elinogrel, ketanserin, relugolix, and sufugolix were included in the following algorithm steps and showed TRPA1 antagonism probabilities of 93.04, 7.30, 99.93, and 43.93%. Neither elinogrel nor relugolix passed the BBB permeability screening. Although melatonin showed a 95.53% flexophore similarity with one TRPA1 inhibitor, the generated prediction algorithm did not retrieve the molecule as a potent TRPA1 antagonist.

A total of 97 drugs were classified as molecules with good BBB permeability and CNS exposure, but only 10 molecules showed TRPA1 inhibition probabilities higher than 90%. Desvenlafaxine (*O*-desmethylvenlafaxine), paliperidone (9-hydroxyrisperidone), and febuxostat were ranked as predicted TRPA1 inhibitors with potential use in MS management. Desvenlafaxine and paliperidone exhibited a predicted binding energy lower than the suggested threshold obtained by analyzing the TRPA1 inhibition probability-predicted binding energy scatter plot. The docking poses and the binding sites analysis of the three molecules indicated that desvenlafaxine may be a promising allosteric channel pore inhibitor by interacting with both A-967079 and HC-030031 binding sites, while paliperidone and febuxostat are prone to induce allosteric inhibition by interacting with the putative HC-030031 binding site.

Recent studies showed that desvenlafaxine is an effective antinociceptive agent in diabetic peripheral neuropathy, hence a potential role of the predicted TRPA1 antagonism could be further investigated in such conditions [82]. Venlafaxine, the parent molecule of desvenlafaxine, showed an inhibition probability of 6.64%, suggesting that the hydrogen bond donor property of the demethylated metabolite could be crucial for receptor inhibition, as shown by the simulated protein–ligand interactions. A recent study reported that venlafaxine suppressed pro-inflammatory cytokines in experimental autoimmune encephalomyelitis (EAE) and alleviated cuprizone-induced demyelination and neuroinflammation, showing promising results in MS murine models [83,84]. Risperidone was also previously shown to exert beneficial effects in the EAE model, significantly reducing microglia and macrophage activation in CNS [85], but the molecule did not satisfy the graph mining score necessary for TRPA1 activity prediction. We can hypothesize that, for both drugs, the effect appears after being metabolized.

Other authors revealed that febuxostat was efficient in managing secondary progressive EAE by restoring mitochondrial ATP production and reducing neurodegeneration [80,81]. These data suggest that the potential TRPA1 inhibitory activity of desvenlafaxine, paliperidone, and febuxostat should be further explored as possible synergistic mechanisms targeting pathology progression in animal models of multiple sclerosis.

Drug repositioning approaches can be often limited by the safety profiles of the screened, approved, or even withdrawn candidates. Literature data show that desvenlafaxine has a safety profile consistent with other antidepressants and a warning was issued regarding the risk of suicidal thinking [75,86]. Paliperidone formulations were proven relatively safe, but were, however, linked to deaths caused by heart disease or infections in adults suffering from dementia-related psychosis [77]. Recent safety data incriminated febuxostat for deaths linked to cardiovascular disease, and the mortality was higher when compared with allopurinol [87,88].

## 5. Conclusions

A step-by-step screening algorithm was implemented in order to identify new potential TRPA1 antagonists and was validated by model quality parameters, showing an excellent prediction accuracy. This method is not limited to this target and could be successful in other in silico studies. The proposed prediction model revealed desvenlafaxine, paliperidone, and febuxostat as potential therapeutic solutions for MS treatment, targeting mainly non-covalent TRPA1 inhibition. Some limitations of the generated model such as descriptor redundancy and overfitting were acknowledged, and the algorithm could be improved by developing machine learning techniques for prospective and similar screening methodologies in the pursuit of other repurposable drug candidates. Future in vitro and in vivo studies are needed to confirm the biological activity of the selected hit molecules and to evaluate the usefulness of pharmacological inhibition of TRPA1 channels in animal models of demyelination.

## Figures and Tables

**Figure 1 pharmaceutics-11-00446-f001:**
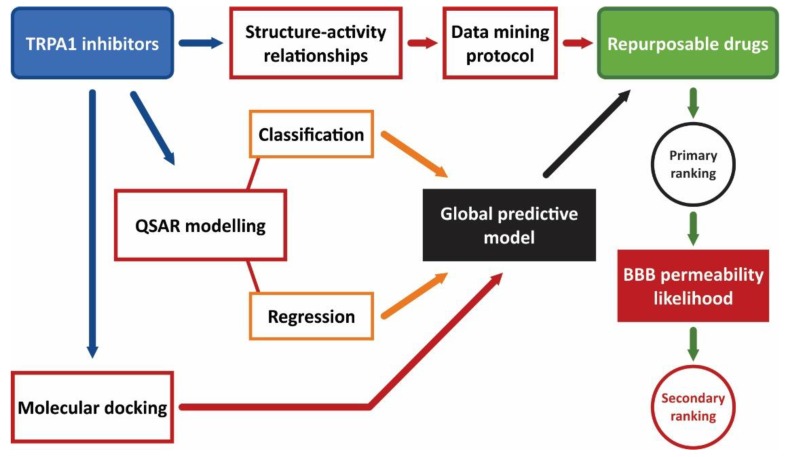
Virtual screening algorithm for discovery of novel potential TRPA1 antagonists.

**Figure 2 pharmaceutics-11-00446-f002:**
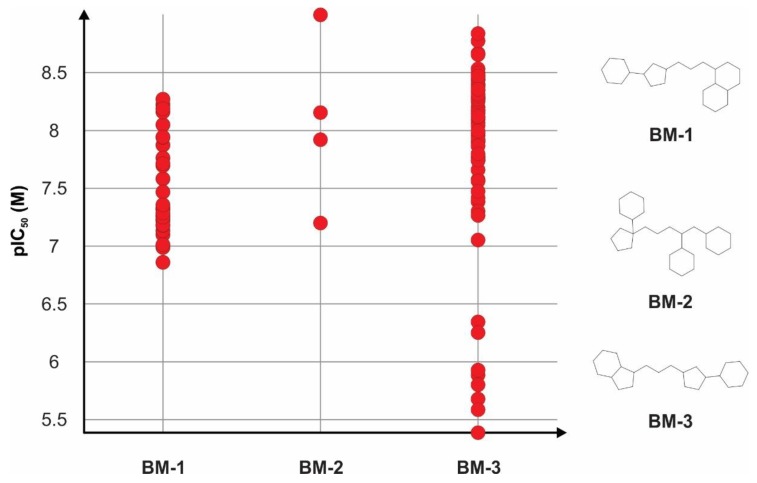
Bemis-Murcko skeletons identified with significant differences in TRPA1 inhibitory activity using statistical comparison of mean pIC_50_ values.

**Figure 3 pharmaceutics-11-00446-f003:**
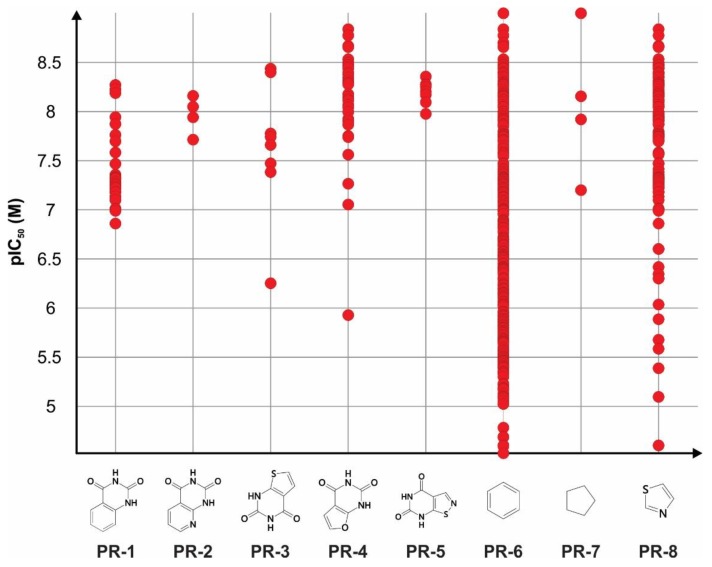
Plain rings contained by compounds with significantly higher TRPA1 inhibitory activity using statistical comparison of mean pIC_50_ (M) values.

**Figure 4 pharmaceutics-11-00446-f004:**
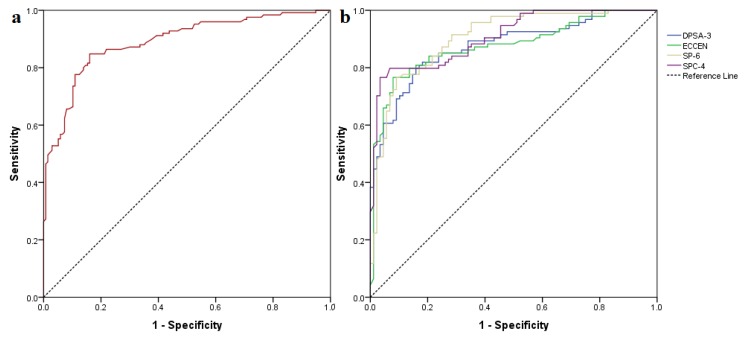
(**a**) global ROC curve of the generated classification model; (**b**) ROC curves of chosen classifiers for the training dataset.

**Figure 5 pharmaceutics-11-00446-f005:**
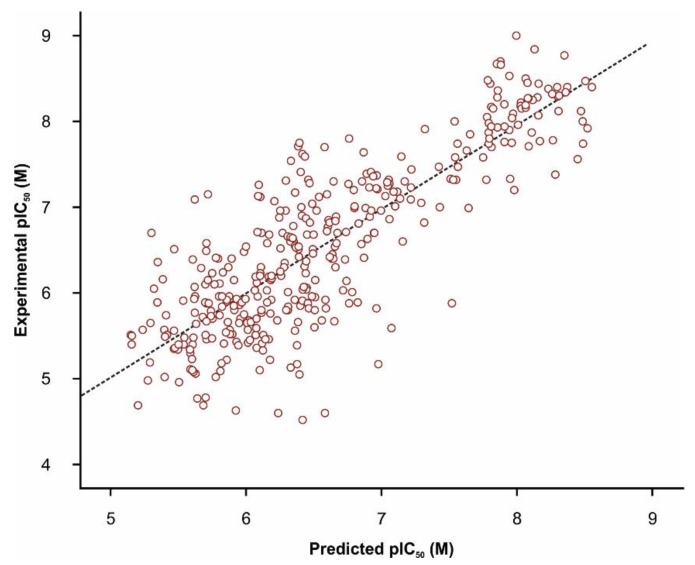
Correlation plot between experimental and MLR predicted activity values (pIC_50_*pred*) of the TRPA1 inhibitor dataset (R^2^*pred* = 0.700).

**Figure 6 pharmaceutics-11-00446-f006:**
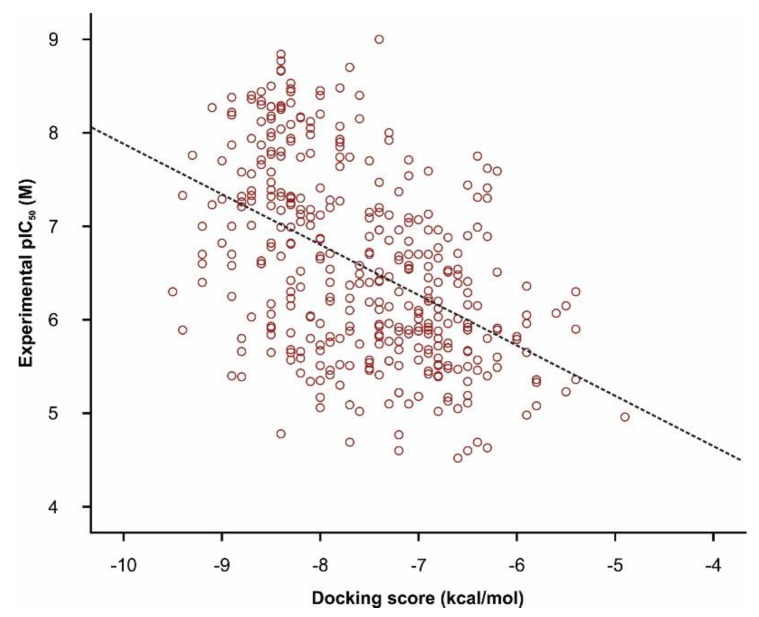
Correlation plot between experimental pIC_50_ (M) and docking scores (kcal/mol) of TRPA1 antagonists (*R*^2^ = 0.226) in molecular docking simulations.

**Figure 7 pharmaceutics-11-00446-f007:**
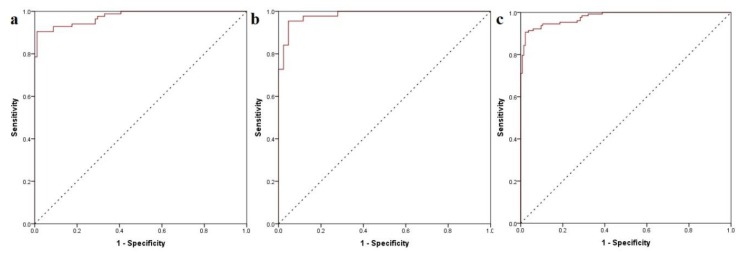
(**a**) ROC curve of calibration (training) subset; (**b**) ROC curve of validation (test) subset; (**c**) global ROC curve of the predictive model.

**Figure 8 pharmaceutics-11-00446-f008:**
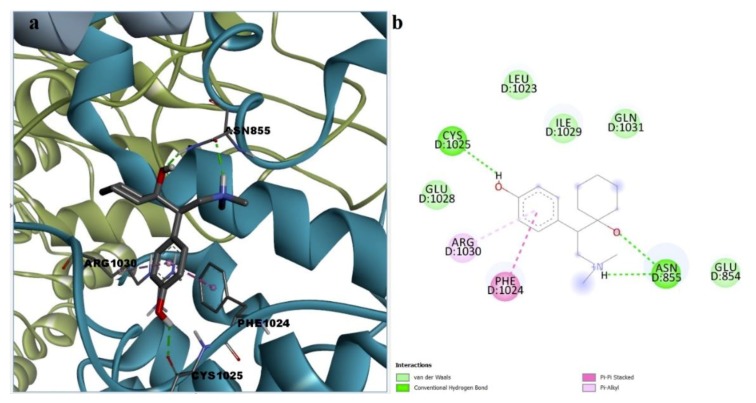
(**a**) 3D binding conformation of desvenlafaxine into putative HC-030031 binding site; (**b**)2D diagram of protein-ligand interactions between TRPA1 and desvenlafaxine.

**Figure 9 pharmaceutics-11-00446-f009:**
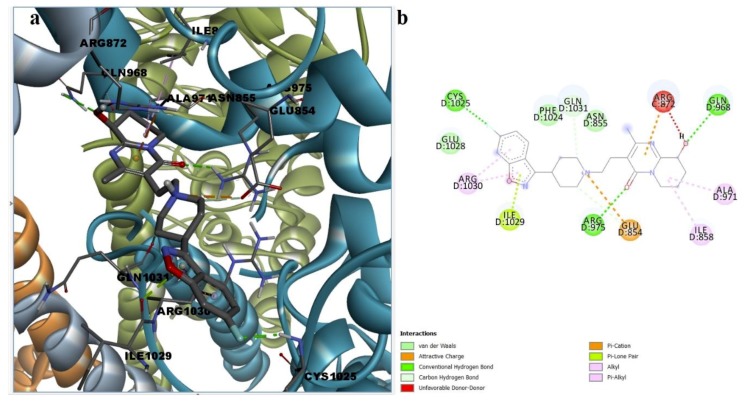
(**a**) 3D binding conformation of paliperidone into putative HC-030031 binding site; (**b**) 2D diagram of protein-ligand interactions between TRPA1 and paliperidone.

**Figure 10 pharmaceutics-11-00446-f010:**
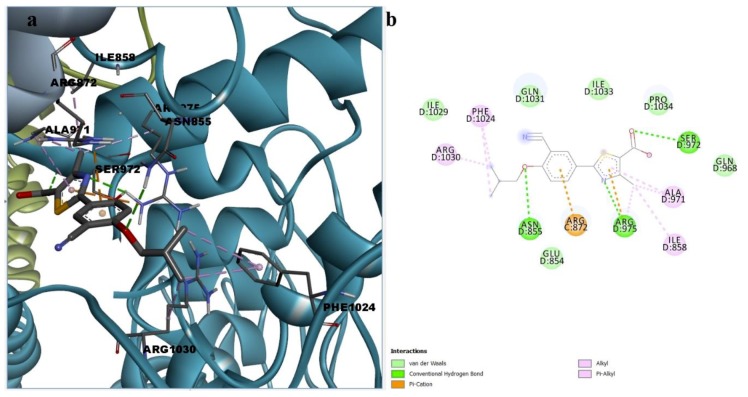
(**a**) 3D binding conformation of febuxostat into putative HC-030031 binding site; (**b**) 2D diagram of protein-ligand interactions between TRPA1 and febuxostat.

**Table 1 pharmaceutics-11-00446-t001:** Structural scaffolds associated with significantly higher biological activity.

Scaffold Type	Identifier	No. of Compounds	Mean ± SD pIC_50_ (M)	Mean Difference (Present − Absent)
Bemis-Murcko skeleton	BM-1	35	7.47 ± 0.39	0.98
	BM-2	4	8.06 ± 0.74	1.50
	BM-3	63	7.78 ± 0.83	1.46
Plain rings	PR-1	31	7.40 ± 0.36	0.90
	PR-2	4	7.96 ± 0.19	1.40
	PR-3	8	7.64 ± 0.68	1.09
	PR-4	33	8.11 ± 0.36	1.69
	PR-5	8	8.20 ± 0.12	1.66
	PR-6	360	6.60 ± 1.00	1.13
	PR-7	4	8.06 ± 0.74	1.50
	PR-8	99	7.59 ± 0.83	1.39
Most central ring	MCR-1	99	7.59 ± 0.83	1.39

SD—standard deviation.

**Table 2 pharmaceutics-11-00446-t002:** Kier-Hall smarts descriptors associated with significantly higher TRPA1 inhibition activity.

Kier-Hall Smarts Descriptor	Identifier	Atom Group	No. of Compounds (Frequency)	Mean ± SD pIC_50_ (M)	Mean Difference (Present – Absent)
khs.ssCH_2_	AG-1	–CH_2_–	248 (66.84%)	6.84 ± 1.02	0.82
khs.dssC	AG-2	=C<	271 (73.04%)	6.67 ± 1.07	0.36
khs.aaaC	AG-3	Ar. C	229 (61.72%)	6.83 ± 1.01	0.67
khs.ssssC	AG-4	>C<	181 (48.78%)	7.00 ± 1.06	0.84
khs.dsN	AG-5	=N–	128 (34.50%)	7.22 ± 1.08	0.99
khs.aaN	AG-6	Ar. N	226 (60.91%)	6.79 ± 1.05	0.55
khs.aasN	AG-7	Ar. N–	173 (46.63%)	7.03 ± 1.00	0.86
khs.sOH	AG-8	–OH	140 (37.73%)	7.23 ± 1.06	1.05
khs.dO	AG-9	=O	271 (73.04%)	6.77 ± 1.01	0.76
khs.aaO	AG-10	Ar. O	63 (16.98%)	7.36 ± 0.99	0.94
khs.sF	AG-11	–F	231 (62.26%)	6.79 ± 1.08	0.58
khs.aaS	AG-12	Ar. S	130 (35.04%)	7.24 ± 1.01	1.02
khs.sCl	AG-13	–Cl	108 (29.11%)	6.78 ± 0.95	0.29

SD—standard deviation; Ar.—aromatic.

**Table 3 pharmaceutics-11-00446-t003:** Number of DrugBank compounds found to feature atom groups and scaffolds previously selected in the structure–activity relationship (SAR) analysis.

Atom Groups		Scaffolds	
Identifier	No. of Structures	Identifier	No. of Structures
AG-1	6388	BM-1	0
AG-2	4962	BM-2	0
AG-3	1857	BM-3	0
AG-4	1805	PR-1	6
AG-5	459	PR-2	0
AG-6	2384	PR-3	2
AG-7	1360	PR-4	0
AG-8	4179	PR-5	0
AG-9	5776	PR-6	2442
AG-10	383	PR-7	21
AG-11	1009	PR-8	82
AG-12	428	MCR-1	46
AG-13	952		

**Table 4 pharmaceutics-11-00446-t004:** Statistics and description of selected classifiers.

Classifier	Description	Cutoff Threshold	Sensibility	Specificity	ROC AUC
DPSA3	difference between charge weighted partial positive surface area and charge weighted partial negative surface area	65.36	0.894	0.659	0.876
ECCEN	eccentric connectivity index	440	0.872	0.636	0.875
SP6	Kier-Hall Chi path index of order 6	3.67	0.957	0.648	0.903
SPC4	Kier-Hall Chi path cluster index of order 4	3.98	0.904	0.602	0.908

* ROC: receiver operating characteristics; AUC: area under the curve.

**Table 5 pharmaceutics-11-00446-t005:** Multiple linear regressions model (MLR) quantitative structure-activity relationship (QSAR) model descriptors and evaluation metrics.

Molecular Descriptors		Model Statistics	
Variable	Description		
GRAV4	gravitational index of all heavy atoms	R^2^	0.707
khs.dO	keto oxygen e-state fragments count	R^2^*pred*	0.681
nHBAcc	hydrogen bond acceptors count	RMSEC	0.455
C2SP3	singly bound carbon bound to two other carbons	RMSEV	0.515
DPSA3	difference between charge weighted partial positive surface area and charge weighted partial negative surface area	Variables	11
khs.ssCH2	–CH_2_– e-state fragments count		
nAcid	acidic groups count		
khs.ddsN	–NO_2_ e-state fragments count		
C2SP2	doubly bound carbon bound to two other carbons		
MDEN13	molecular distance edge between all primary and tertiary nitrogen atoms		
C3SP2	doubly bound carbon bound to three other carbons		

*R*^2^—squared correlation coefficient of the training subset: R^2^*pred*—squared correlation coefficient of the predicted (test) subset; RMSEC—Root Mean Square Error of the calibration dataset (training subset); RMSEV—Root Mean Square Error of the validation dataset (test subset).

**Table 6 pharmaceutics-11-00446-t006:** Top 10 secondarily ranked potential TRPA1 inhibitors with good central nervous system (CNS) exposure based on the binary logistic regression equation used as a global prediction model and blood brain barrier (BBB) permeation filtering.

DrugBank ID	Generic name	Drug groups	Biological activity	Score	Activity class	pIC_50_*pred* (M)	Δ*G* (kcal/mol)	P
DB11629	Laropiprant	A, I, W	selective DP1 antagonist	3	1	9.38	−7.3	1.00000
DB11644	Tafamidis	A, I	TTR dissociation inhibitor	3	1	9.01	−6.8	0.99999
DB06700	Desvenlafaxine	A, I	SNRI	3	1	8.17	−7.9	0.99976
DB01267	Paliperidone	A	antipsychotic	3	1	7.39	−9.0	0.99661
DB04854	Febuxostat	A	XO inhibitor	3	1	7.28	−6.4	0.98517
DB02266	Flufenamic Acid	A	NSAID	3	1	7.13	−7.1	0.98032
DB00957	Norgestimate	A, I	sex hormone	3	0	7.76	−6.9	0.97359
DB04908	Flibanserin	A, I	5-HTA1/2 agonist/antagonist	4	0	7.59	−8.3	0.96989
DB01600	Tiaprofenic acid	A	NSAID	3	0	7.65	−6.3	0.94887
DB01359	Penbutolol	A, I	beta-blocker	3	1	7.00	−5.7	0.94310

Score—data mining score; pIC_50_*pred* (M)—MLR predicted pIC_50_; Δ*G*—predicted binding energy (kcal/mol); P—probability of TRPA1 inhibition; I—investigational; A—approved; W—withdrawn; TTR—transthyretin; SNRI—serotonin-norepinephrine reuptake inhibitor; XO—xanthine oxidase; NSAID—nonsteroidal anti-inflammatory drug.

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
