# Peer review of "Computational Drug Repurposing Algorithm Targeting TRPA1 Calcium Channel as a Potential Therapeutic Solution for Multiple Sclerosis"

_pharmaceutics, 2019, doi:10.3390/pharmaceutics11090446_

Round 1

Reviewer 1 Report

see attached.

Author Response

We would like to thank all the reviewers for the time they have taken to review our manuscript.  All the corrections were made using “Track Changes” as suggested and detailed point-by-point responses to each of the comments raised by the two reviewers are also provided as follows.

Reviewer 1

The manuscript is interesting and the authors used public databases and computational methods to identify potential drug candidates to be repurposed for treating multiple sclerosis. A total of 86 drugs were classified as molecules with good BBB permeability and CNS exposure, Comments.

Based on the approved label, the absorption of radiolabeled febuxostat following oral dose administration was estimated to be at least 49%. That is febuxostat is not well absorbed across intestinal epithelium, suggesting that the amount across the BBB after oral febuxostat will be even lower. Please add information of the extent of oral absorption of desvenlafaxine, paliperidone and febuxostat to provide relevant context for those 3 compounds.

R: As recommended, absolute bioavailability data for all 3 drugs have been added in the results section. Other reports suggest that febuxostat has a bioavailability of 84%. However, febuxostat was proven efficient in ameliorating murine experimental autoimmune encephalomyelitis, therefore endorsing that febuxostat is capable of relevant BBB diffusion.

The recent box warning regarding cardiovascular death related to use of febuxostat should be added to the manuscript. Since this manuscript is a mining and computational manuscript, adding safety information and absorption information of the three candidates is needed.

R: Safety warnings have been added in discussion section for all 3 molecules.

Line 29: Please revise the word “efficacy” from the sentence of the algorithm generated for screening TRPA1 inhibitors shows efficacy in identifying ligands, to reflect the fact that algorithm was used and the ligands have not been tested yet for their in-vitro potency or preclinical effectiveness.

R: The part regarding the efficacy of the algorithm was removed from the phrase.

In the Binary classification model section, please provide the rationale for the PIC50 cutoff to divide compounds into 2 groups (lines 168-169). Though the authors cited a reference, the therapeutic effects in the cited reference and this manuscript are different. It is not clear how and why pIC50 values < 6 6 and with pIC50 > 7 were chosen.

R: The rationale behind using the mentioned cutoffs was added in the manuscript and a reference was cited accordingly, as suggested. The reference mentioned by the reviewer is related to the cutoff classification model which was used in both manuscripts, the method being independent on the biological target (now lines 182-185).

The method for structural similarity comparison was not described in the manuscript. Details and references are needed.

R: Details and references regarding structural similarity comparison have been provided in the manuscript, as indicated.

Line 174: please define “variables”

R: The word “variables” was defined as suggested (now line 188).

The introduction was expanded as indicated in the reviewer’s report. Additionally, some of the figures were replaced after font editing in order to homogenize the aspect of the paper. Figure 8 (now Figure 5) was replaced because it was accidentally inserted and it reported a correlation plot from another QSAR model obtained for TRPA1 inhibitors, but not the one selected for further screening.

Reviewer 2 Report

Recommendation: Reconsider after major revisions noted.

Researchers used computational drug discovery and data mining methods, performed an in silico screening study combining chemical graph mining, QSAR modeling and molecular docking techniques to identify repurposable drugs as potent TRPA1 antagonists that may serve as potential treatments for Multiple Sclerosis patients. Overall the manuscript is well written and clear. However, the way they used molecular docking was not adequate, nor were the key residues chosen for site-directing docking simulations. As a result, this makes the results obtained through this methodology unreliable. I would recommend its publication after major  revisions.

1)     Why were the properties of compounds coming from the drugbank recalculated? Those properties were already included in that database; therefore, performing those calculations again would undoubtedly lead to errors.

2)     For the ligands, were the charges kept during the parametrization process before performing molecular docking?

3)     In Figure 1, you did not include the “Compounds serving for drug repurposing screening.”

4)     Line 136: Change “In order to establish the key structural” by “To establish the key physicochemical.”

5)     Line 142: What do you mean by “center of the molecule,” is it the mass center of the molecule? Please clarify.

6)     Line 198. Change “crystal structure of human TRPA1 (PDB: 3j9p)” by “crystal structure of human TRPA1 (PDB code: 3J9P)”

7)     The problem of the human TRPA1 crystal structure’s low resolution (4.24) (PDB code: 3J9P) was not described and tackled. This is a significant problem since sidechain placements depend on the resolution, and the molecular docking algorithm used in this work is not optimal for this type of structures. Structures with low resolution present evident problems when representing how certain chemical compounds may or may not interact with such structures.

I recommend visualizing the electron density map (EDM) from the Uppsala, Sweden electron density server. Check the EDM in TRPA1´s binding site(s), and see if the EDM supports the side chain placement. Often, in low-resolution structures, the crystallographer will place side chains in reasonable conformations without the side chain electron density. This information is reflected in B-Factors as well. Together, B-factors and by-eye analysis of the EDM should give a better idea of the accuracy of the binding site's crystal structure than the overall resolution alone.

A methodology such as the induced-fit docking should be chosen to minimize such errors. This methodology not only allows the selected compounds to be coupled, but also takes into account the flexibility of the receptor, allowing it to adapt to the ligand, obtaining more precise results. Other approximations to re-score and improve the initial docking results such as MM-(GB/PB)-SA have proven to be useful for these types of problems as well.

Additionally, the binding sites were incorrectly chosen. A "blind" docking was done without considering the reported residues forming the binding site(s) of these type of ion channels. The results reported here may not necessarily be an accurate representation of how these ligands interact with this ion channel. It is widely documented that the success of molecular docking, among other things, lies in how the binding site residues are chosen, and, in this work, a site was not chosen logically since it did not consider all the drugbank information which indicates which TRPA1 residues interact with the TRPA1 blockers reported in the literature. Those residues must be considered since they guarantee the evaluation of new simulations in the sites with catalytic activity within the channel.

Finally, the authors chose the best solution (top-1) according to the final docking score. It is well documented that the best pose according to the autodock and autodock vina score function (among other software) is not necessarily the one corresponding to reality (Int. J. Mol. Sci. 2016, 17, 525; Molecules 2018, 23(5) 1038); for this reason, you should consider the solutions suggested (top-5) and analyze deeper to observe which is the best pose (with mainly experimental information), which is not necessarily related to the kcal/mol value reported by the software.

8)     The concentration units must be indicated for IC50 and pIC50 (in tables and throughout the manuscript).

9)     The kilo prefix is k and not K, according to the International System of Units. Please use kcal/mol instead of Kcal/mol throughout the manuscript.

10)   Refer to docking score in kcal/mol instead Binding free energy or pKi

11)   Docking results. The molecular docking results do not predict the free binding energies of the protein-ligand complexes; this statement is wrong. It is well documented that to predict relative free binding energies (dG) more robust and accurate methods such as MM-GB/PB-SA, FEP, Molecular dynamics simulations, or Quantum mechanics must be used. The dG energy is thermodynamically defined as dG = dH – TdS. Authors need to demonstrate that the docking method employed in this work was used to calculate the enthalpic and entropic changes; otherwise, they would have to use other methods to calculate the relative dG, or be aware of the docking methodology scope, and discuss the results according to the scope of the chosen method.

12)   Figure 9. Please correlate the experimental pIC50 (M) with the docking score (kcal/mol). The pKipred is not an accurate prediction and must not be considered in the present study.

13)   There are too many figures (15) and tables (12). Please consider to include the most important tables and figures in the main manuscript. The remaining ones should be reported as supplementary material.

14)   Fig 13, 14, and 15. Please use either yellow dotted lines or black dotted lines to represent H-Bonds, but not both.

15)   Fig 14. In the Fig14a the H-bond is missing.

16)   There are mistakes in the reported chemical structures of desvenlafaxine, paliperidone, and febuxostat. In Fig 12, the desvenlafaxine has a methoxy moiety, and in Fig13a-b this moiety is a -OH. In Fig 12, the paliperidone has a methyl moiety, and in Fig 14a-b, this moiety is a -OH. Besides, the paliperidone structure is not consistent in the reported results.

17)   In Fig 12, all N atoms were not charged. Furthermore, in Fig13a, the amino is not charged, and in fig13b is charged. The same mistake was identified in Fig14a and b regarding Fig12. In Fig12 and Fig15b the febuxostat is not charged, but in Fig15a it is protonated. All of these results are inconsistent and clearly raise many doubts as to how the structures were prepared for this study. Validation must be provided to be sure about the atoms partial and total charge of the studied structures, as well as for the 3D structures.

Author Response

We would like to thank all the reviewers for the time they have taken to review our manuscript.  All the corrections were made using “Track Changes” as suggested and detailed point-by-point responses to each of the comments raised by the two reviewers are also provided as follows.

Reviewer 2

Researchers used computational drug discovery and data mining methods, performed an in silico screening study combining chemical graph mining, QSAR modeling and molecular docking techniques to identify repurposable drugs as potent TRPA1 antagonists that may serve as potential treatments for Multiple Sclerosis patients. Overall the manuscript is well written and clear. However, the way they used molecular docking was not adequate, nor were the key residues chosen for site-directing docking simulations. As a result, this makes the results obtained through this methodology unreliable. I would recommend its publication after major  revisions.

1)     Why were the properties of compounds coming from the drugbank recalculated? Those properties were already included in that database; therefore, performing those calculations again would undoubtedly lead to errors.

R: Properties coming from drugbank database were recalculated in order to have descriptors coming from the same source for both TRPA1 inhibitors (training dataset) and repurposable drugs, in order to generate uniform pools of molecular descriptors, calculated using the same prediction algorithm.

2)     For the ligands, were the charges kept during the parametrization process before performing molecular docking?

R: Charges/protonation corresponding to pH = 7.4 were kept for the docking protocol.

3)     In Figure 1, you did not include the “Compounds serving for drug repurposing screening.”

R:  In figure 1, there is a green box labeled “Repurposing drugs” corresponding to “Compounds serving for drug repurposing screening”.

4)     Line 136: Change “In order to establish the key structural” by “To establish the key physicochemical.”

R:  The phrase was changed accordingly. (now line 142)

5)     Line 142: What do you mean by “center of the molecule,” is it the mass center of the molecule? Please clarify.

R: Center of the molecule in this case refers to the graph center. The notion was clarified in the manuscript as suggested. (now line 148)

6)     Line 198. Change “crystal structure of human TRPA1 (PDB: 3j9p)” by “crystal structure of human TRPA1 (PDB code: 3J9P)”

R: Changes were made accordingly.

7)     The problem of the human TRPA1 crystal structure’s low resolution (4.24) (PDB code: 3J9P) was not described and tackled. This is a significant problem since sidechain placements depend on the resolution, and the molecular docking algorithm used in this work is not optimal for this type of structures. Structures with low resolution present evident problems when representing how certain chemical compounds may or may not interact with such structures.

I recommend visualizing the electron density map (EDM) from the Uppsala, Sweden electron density server. Check the EDM in TRPA1´s binding site(s), and see if the EDM supports the side chain placement. Often, in low-resolution structures, the crystallographer will place side chains in reasonable conformations without the side chain electron density. This information is reflected in B-Factors as well. Together, B-factors and by-eye analysis of the EDM should give a better idea of the accuracy of the binding site's crystal structure than the overall resolution alone.

 R: The reviewer described accurately with strong scientific arguments the problem of the studied protein. Indeed, TRPA1 cryoEM structure has a low resolution and large B-factors. Thus, in order to minimize the docking inaccuracy due to the faulted protein, the molecular docking protocol was improved. In order to optimize the protein structure, missing side chains and loops were added and refined, the structure was energetically minimized and the docking simulations were redone.

A methodology such as the induced-fit docking should be chosen to minimize such errors. This methodology not only allows the selected compounds to be coupled, but also takes into account the flexibility of the receptor, allowing it to adapt to the ligand, obtaining more precise results. Other approximations to re-score and improve the initial docking results such as MM-(GB/PB)-SA have proven to be useful for these types of problems as well.

 R: We acknowledge the increase in fidelity of the methodology described by the reviewer. In our project we intended to perform a high throughput virtual screening, and docking all ligands with an induced fit method would be computationally expensive. Usually, docking a large compound library can be done using less precise methods, in order to shorten the otherwise very long simulation times. For instance, Schrödinger’s Glide docking module can use three types of docking algorithms: low precision for virtual screening, standard precision and extra precision. Moreover, AutoDock Vina required manual selection of flexible sidechains, requiring the researcher to precisely know all the residues that can suffer conformational changes due to ligand binding. Thus, we proposed a middle ground solution, by docking the two gold standard TRPA1 inhibitors, A-967079 and HC-030031 into their assumed binding sites using the induced fit approach, by manually selecting flexible residues until the ligands fit into an optimal conformation. Thereafter, the coordinates of the protein were updated with the new side chain coordinates, as a mean to mimic the conformational structure of pharmacologically inhibited TRPA1. Rescoring with molecular mechanics/dynamics approaches such as MM-(GB/PB)-SA was reported to yield good results, however, it is hard to apply to virtual screening of a large dataset considering these types of simulations require many computational resources.

Additionally, the binding sites were incorrectly chosen. A "blind" docking was done without considering the reported residues forming the binding site(s) of these type of ion channels. The results reported here may not necessarily be an accurate representation of how these ligands interact with this ion channel. It is widely documented that the success of molecular docking, among other things, lies in how the binding site residues are chosen, and, in this work, a site was not chosen logically since it did not consider all the drugbank information which indicates which TRPA1 residues interact with the TRPA1 blockers reported in the literature. Those residues must be considered since they guarantee the evaluation of new simulations in the sites with catalytic activity within the channel.

 R: Our first intention was to use a blind docking approach in order to include all the druggable cavities in the protein, since there are already two known different binding sites. However, the docking protocol was changed and a directed approach was used, by setting the searching space to include the two reported binding sites of other known antagonists.

Finally, the authors chose the best solution (top-1) according to the final docking score. It is well documented that the best pose according to the autodock and autodock vina score function (among other software) is not necessarily the one corresponding to reality (Int. J. Mol. Sci. 2016, 17, 525; Molecules 2018, 23(5) 1038); for this reason, you should consider the solutions suggested (top-5) and analyze deeper to observe which is the best pose (with mainly experimental information), which is not necessarily related to the kcal/mol value reported by the software.

R: Top 5 poses were explored in the redone docking experiment for A-967079 and HC-030031 in order to identify the most suitable protein-ligand conformation for further optimization. Moreover, top 5 poses were also explored for the 3 proposed hit compounds. However, other poses are hard to explore during the virtual screening process, since it would require to manually analyze approximately 6000 docking poses, which is not befitted, but can be visualized for the final hit structures.

8)     The concentration units must be indicated for IC50 and pIC50 (in tables and throughout the manuscript).

R: Concentration units have been indicated for both IC50 and pIC50 as suggested.

9)     The kilo prefix is k and not K, according to the International System of Units. Please use kcal/mol instead of Kcal/mol throughout the manuscript.

R:  Changes were made accordingly.

10)   Refer to docking score in kcal/mol instead Binding free energy or pKi

R: Changes were made in the text.

11)   Docking results. The molecular docking results do not predict the free binding energies of the protein-ligand complexes; this statement is wrong. It is well documented that to predict relative free binding energies (dG) more robust and accurate methods such as MM-GB/PB-SA, FEP, Molecular dynamics simulations, or Quantum mechanics must be used. The dG energy is thermodynamically defined as dG = dH – TdS. Authors need to demonstrate that the docking method employed in this work was used to calculate the enthalpic and entropic changes; otherwise, they would have to use other methods to calculate the relative dG, or be aware of the docking methodology scope, and discuss the results according to the scope of the chosen method.

R: Indeed, in the manuscript there was a phrase referring to the free binding energy of the complex and it was corrected. However, the developers of AutoDock Vina claim in their user manual and published paper that the docking score is represented by the binding affinity or binding energy of the ligand, thus we referred to the docking results as docking score, binding affinity or binding energy.

12)   Figure 9. Please correlate the experimental pIC50 (M) with the docking score (kcal/mol). The pKipred is not an accurate prediction and must not be considered in the present study.

R: pKi pred is proportional to the docking score and was calculated using the formula reported in the AutoDock Vina manual. However, pKi values were removed from the manuscript to ease the interpretation of the results and the suggested change was made.

13)   There are too many figures (15) and tables (12). Please consider to include the most important tables and figures in the main manuscript. The remaining ones should be reported as supplementary material.

R: Several figures and tables were reported in supplementary materials as recommended.

14)   Fig 13, 14, and 15. Please use either yellow dotted lines or black dotted lines to represent H-Bonds, but not both.

R: Figures were replaced.

15)   Fig 14. In the Fig14a the H-bond is missing.

R: Figure was replaced.

16)   There are mistakes in the reported chemical structures of desvenlafaxine, paliperidone, and febuxostat. In Fig 12, the desvenlafaxine has a methoxy moiety, and in Fig13a-b this moiety is a -OH. In Fig 12, the paliperidone has a methyl moiety, and in Fig 14a-b, this moiety is a -OH. Besides, the paliperidone structure is not consistent in the reported results.

R: Figures were corrected as noted

17)   In Fig 12, all N atoms were not charged. Furthermore, in Fig13a, the amino is not charged, and in fig13b is charged. The same mistake was identified in Fig14a and b regarding Fig12. In Fig12 and Fig15b the febuxostat is not charged, but in Fig15a it is protonated. All of these results are inconsistent and clearly raise many doubts as to how the structures were prepared for this study. Validation must be provided to be sure about the atoms partial and total charge of the studied structures, as well as for the 3D structures.

R: Figures were replaced to provide more consistency.

Additionally, some of the figures were replaced after font editing in order to homogenize the aspect of the paper. Figure 8 (now Figure 5) was replaced because it was accidentally inserted and it reported a correlation plot from another QSAR model obtained for TRPA1 inhibitors, but not the one selected for further screening.

Round 2

Reviewer 2 Report

Recommendation: Reconsider after minor revisions noted.

Line 215: Again, change “RCSB Protein Data Bank (PDB: 3J9P)” by “RCSB Protein Data Bank (PDB code: 3J9P)” Line 224: The authors stated they performed “molecular dynamics simulations”. In their work they performed “molecular docking simulations”, which is a different technique. Please correct. Line 257: Change “and the predicted pIC50 (pIC50pred) and binding energy (ΔG)” by “the predicted pIC50 (pIC50pred), and binding energy (ΔG)” Change “binding energy” or “binding affinity” by “predicted binding energy” or “predicted binding affinity”. AutoDock Vina Manual and published paper (Trott, O., & Olson, A. J. (2010). Journal of computational chemistry, 31(2), 455-461.) refer to the docking score as predicted binding energy.

Author Response

We would like to thank the reviewer for all the efforts to improve our article.

Line 215: Again, change “RCSB Protein Data Bank (PDB: 3J9P)” by “RCSB Protein Data Bank (PDB code: 3J9P)”

R: Line 215 has been changed as suggested.

Line 224: The authors stated they performed “molecular dynamics simulations”. In their work they performed “molecular docking simulations”, which is a different technique.

R: We apologize for the confusion created. The authors wanted to state that analyses performed by other authors in their respective papers regarding electron density maps and molecular dynamics simulations indicated that the two inhibitors block the channel via an induced fit mechanism. Thus, the noted phrase was changed to “Previously reported analyses of electron density maps and molecular dynamics simulations indicated that inhibitors A-967079 and HC-030031 bind to TRPA1 and interfere with channel gating via an induced fit mechanism”.

Please correct. Line 257: Change “and the predicted pIC50 (pIC50pred) and binding energy (ΔG)” by “the predicted pIC50 (pIC50pred), and binding energy (ΔG)”

R: Corrections have been made as indicated.

Change “binding energy” or “binding affinity” by “predicted binding energy” or “predicted binding affinity”. AutoDock Vina Manual and published paper (Trott, O., & Olson, A. J. (2010). Journal of computational chemistry, 31(2), 455-461.) refer to the docking score as predicted binding energy

R: Changes have been made accordingly.